# Analysis of the Influence of Pre-Pregnancy BMI and Weight Gain during Pregnancy on the Weight of Healthy Children during the First 2 Years of Life: A Prospective Study

**DOI:** 10.3390/children9101431

**Published:** 2022-09-21

**Authors:** Beata Łoniewska, Kaja Michalczyk, Konrad Podsiadło, Karolina Adamek, Barbara Michalczyk, Piotr Tousty, Mariusz Kaczmarczyk, Igor Łoniewski

**Affiliations:** 1Department of Neonatology and Intensive Neonatal Care, Pomeranian Medical University in Szczecin, al. Powstańców Wielkopolskich 72, 70-111 Szczecin, Poland; 2Department of Gynecological Surgery and Gynecological Oncology of Adults and Adolescents, Pomeranian Medical University in Szczecin, al. Powstańców Wielkopolskich 72, 70-111 Szczecin, Poland; 3Department of Clinical and Molecular Biochemistry, Pomeranian Medical University in Szczecin, al. Powstańców Wielkopolskich 72, 70-111 Szczecin, Poland; 4Department of Obstetrics and Gynecology, Pomeranian Medical University in Szczecin, al. Powstańców Wielkopolskich 72, 70-111 Szczecin, Poland; 5Department of Human Nutrition and Metabolomics, Pomeranian Medical University in Szczecin, 71-460 Szczecin, Poland; 6Department of Biochemical Sciences, Pomeranian Medical University in Szczecin, Broniewskiego 24, 71-460 Szczecin, Poland

**Keywords:** prenatal obesity, maternal weight, gestational weight gain, birthweight, childhood obesity

## Abstract

Background: Increased pre-pregnancy maternal BMI (pBMI) and gestational weight gain (GWG) have been found to increase infants’ birthweight and result in the programming of child weight and impact its later weight gain. Aim: To assess the impact of pBMI and GWG on the weight of children from birth to 2 years of age and over the duration of breastfeeding. Methods: Single Centre observational prospective longitudinal cohort study. Data were collected from medical records, and medical history. The analysis of multiple linear and mixed models was involved. Findings: 20% of females were overweight, while 13% were obese before the pregnancy. An overall model, including gender and smoking, indicated a significant impact of pBMI category on a child’s birth mass (*p* = 0.01). The GWG category affected a child’s birth weight (*p* = 0.018, Effect size 0.41). pBMI did not affect the breastfeeding duration. Conclusion: pBMI and GWG correlate with birth weight and weight in neonatal period, however they become insignificant in later childhood. Weight assessment methods among children aged up to two years of age require standardization. Maternal weight before the pregnancy nor the weight gain during the pregnancy do not influence the length of breastfeeding. The biggest limitation was the small sample size and the failure to account for weight gain per trimester of pregnancy. Further research on a larger population should be continued.

## 1. Introduction

The population of overweight and obese women of reproductive age is constantly rising. Pre-pregnancy maternal overweight and obesity and gestational weight gain (GWG) have been demonstrated to increase complication rates during gestation and delivery including gestational hypertension, preeclampsia, gestational diabetes mellitus, increased rates of cesarean delivery, prematurity, still birth, congenital anomalies, macrosomia and shoulder dystocia [1,2,3]. Pre-pregnancy maternal BMI (pBMI) and GWG are associated in a direct manner with infants’ birthweight [4,5] and result in the programming of child weight and impact its later weight gain (“fetal programming”) [6]. The above-mentioned factors have also been shown to affect childhood development causing future obesity, diabetes, neurodevelopment and augmented cardiovascular risk [7,8,9,10]. pBMI and GWG were also demonstrated to have a negative influence on breastfeeding. It is probably caused by physiological, psychosocial and health-related factors [11]. Increased body mass may adversely affect lactation in a direct manner as well as indirectly causing pregnancy and infancy complications that affect breastfeeding [12].

Lifestyle modifications (modification of diet and physical activity, together with behavioral and social support strategies) during pregnancy to reduce excess GWG and pBMI at conception can have a broad range of short- and long-lasting benefits for mothers and infant’s health outcomes [13]. This is why it is of high importance to conduct prospective studies that analyze the relationship between pBMI, GWG and the body weight of children in their initial period of life. The number of such studies is limited as most research has concentrated on the perinatal outcomes and only few have continued in infancy [1,6,8,10,13,14,15,16]. In addition, there are several observations (including a meta-analysis) involving the effect of pBMI on children’s body weight later in life [6,17]. Given the nutritional importance of breastfeeding and the health benefits for both mother and baby [18,19], studies of the effects of maternal body weight on breastfeeding are crucial and necessary. Moreover, studies that evaluate the influence of pBMI on the body weight of children and breastfeeding in early life in the Polish population are sparse and limited to the neonatal period [20]. According to official data from Polish Central Statistical Office, between 2009 and 2019, the percentage of overweight (BMI 25–29.9) among women aged 20–39 increased from 25.8 to 31.3 [21]. This means that more than 30% of women of childbearing age are overweight. This is therefore a significant social and health problem. In Poland, there is no central medical data collection system, so epidemiological studies are based on data obtained from individual cohorts. There are also no prospective studies covering the first years of life of healthy children.

Our specific aims were to assess the impact of pBMI and GWG on the weight of children and breastfeeding duration from birth to 2 years of age. Our hypothesis was that higher pBMI and GWG would be associated with higher birthweight, future infants’ body weight, the prevalence of obesity and shorter breastfeeding.

## 2. Materials and Methods

We used an observational prospective longitudinal cohort study to measure the infants’ weight during the first 24 months of life. For the purpose of the study, we selected 100 mother-child dyads of healthy, full-term newborns from singleton pregnancies born at the Department of Obstetrics, Gynecology and Neonatology. We calculated the minimum sample size (*p* = 0.8, α = 0.05) for Cohen’s f^2 small, medium and large effect size. The calculated minimum sample size to detect small effects was 61 subjects. The study was part of a project that aimed to analyze the intestinal barrier status of mothers before the delivery and babies from birth to the age of two [22,23]. The selection and recruitment of patients was conducted by an obstetrician or/and a neonatologist. All of the patients have given formal consent to participate in the study. The consent to perform the study was obtained from the Bioethics Committee; the study followed the Declaration of Helsinki (2013).

The condition of each newborn after the birth was assessed according to the Apgar scale and on the basis of the results of umbilical-cord blood-gas assessments. The inclusion criteria: healthy, term newborns born to healthy mothers in good condition; all qualified newborns after birth were rated above 7 points on the Apgar scale after 3 min of life, and the pH of the umbilical cord blood was >7.2.

The study excluded children born to mothers with autoimmune diseases (including type 1 diabetes), pregnancies complicated by gestational diabetes mellitus (GDM), with HELLP syndrome (Haemolytic anemia, Elevated Liver enzymes, Low Platelet count), prematurely born (before 37 weeks of gestation), in asphyxia (Apgar ≤ 7 after 3 min of age), with congenital infection (clinical signs of infection, increased CRP, IL6) or with birth defects. Mothers who had a history of infection during pregnancy were not excluded from the study.

Clinical data on maternal weight and height and weight gain during pregnancy, antibiotic treatment during pregnancy, *Streptococcus agalactiae* (GBs) vaginal colonization and type of delivery were collected from medical records, and duration of breastfeeding and cigarette smoking were collected based on the medical history. Depending on pBMI, mothers were categorized into underweight (BMI < 18.5), overweight (BMI 25–29.9), obese (BMI ≥ 30) and normal weight (BMI 18.5–24.9) women [24]. With regard to weight gain during pregnancy, based on centile grids published by Santos et al., (2018), women with inadequate, normal, and excessive weight gain were distinguished [25].

Newborns were eligible for the study based on a physical examination on the first day of life. The mode of delivery and birth weight was noted. Then at 1, 6, 12 and 24 months old, the authors contacted the parents of children to perform interviews about the type of baby’s feeding (naturally vs. artificially) and child’s weight (recorded from the Health Booklet). All measurements of the infants’ weight were referred to girls’ and boys’ charts created by the WHO (The WHO Child Growth Standards) [26]. In accordance with WHO recommendation the prevalence of underweight (<−2 Z-scores), overweight (2–3 Z-scores), and obesity (>3 Z-scores) was determined for children at 12 and 24 months of age and was measured by Z-score. The prevalence of underweight and obesity in the study population of children was also checked according to the type of centile charts used: percentiles weight-for-age and BMI-for-age (Table 1). Analyses of occurrence of abnormal body weight in children according to the maternal pBMI category were based on all classifications.

The breastfeeding period took into account exclusive and partial breastfeeding times. Natural feeding is the feeding of breast milk—either directly from the breast or pumped into a bottle. Artificial feeding involves giving the baby a milk mixture.

Antibiotic therapy was recorded based on the Pregnancy Chart and the history collected from the mother. Detailed information on the doses used and the length of therapy was not available.

### Statistical Analysis

A descriptive analysis was performed, calculating mean, standard deviation for quantitative variables and proportions for qualitative variables. We conducted a correlation matrix analysis calculating Spearman’s R, first among all quantitative variables, then separately for the child’s mass in different time points with the mother’s pBMI. Multiple Linear Models were used to compare the differences in child’s body mass according to different categories of maternal pBMI and GWG, adjusted by smoking, gender, breastfeeding and delivery type, because all these factors can affect children’s body weight [27,28,29,30]. Cohen’s f2 was used as an effect size for multiple linear models. Chi^2^ test was performed to analyze the differences between the proportions of obesity and underweight among different pBMI and GWG categories. Analyses of child’s weight gain over time among the maternal pBMI group was conducted using the general mixed model with gender adjustment. All analysis were performed in the R environment (R v. 4.0.5). A language and environment for statistical computing. R Foundation for Statistical Computing, Vienna, Austria. ISBN 3-900051-07-0, URL http://www.R-project.org/. The FDR correction was applied to *p* values and a threshold of 0.05 statistical significance was used.

## 3. Results

### 3.1. Group Characteristics

During the study period, 27 mother-infant dyads withdrew from the study. The most common cause of withdrawal was the inability to contact the patient or lack of reliable maternal or child anthropometric data. The group characteristics are listed in Table 2. The children group consisted of 44 males and 39 females, of whom 52 were delivered via C-section. 18 mothers smoked cigarettes either before and/or during pregnancy. The mean value of pBMI was 24.06 (standard deviation (sd) = 5.1), before labor 29.68 (sd = 4.73) and mean BMI gain was 5.62 (sd = 2.49). The mean value of child’s birth mass was 3.44 (sd = 0.45), 1st month mass 4.5 (sd = 0.6), 6th month mass 7.93 (sd = 1.09), 12th month mass 10.11 (sd = 1.09) and 24th mass (sd = 1.67). The mothers who had normal pBMI had the highest mass gain, whereas mothers who were obese had the lowest.

All children were breastfed at 1 week of age, 81.5% at 1 month, 51.2% at 6 months, 16.7% at 12 months, and 4.1% at 24 months. pBMI did not affect the length of breastfeeding. In the study, we found no differences in the percentage of women breastfeeding for 6 months depending on the pBMI (Table 3). The differences in the prevalence of cigarette smoking among mothers with different nutritional status before pregnancy were not statistically significant. There were no differences in the prevalence of *Streptococcus agalactiae* (GBs) vaginal colonization and use of antibiotic therapy during pregnancy and delivery (ATB) between women with different pBMI.

### 3.2. Analysis of Children’s Body Mass

The differences in child’s body mass according to different categories of maternal BMI were adjusted by smoking, gender, breastfeeding and delivery type (in case of birth weight only by smoking and gender). The Spearman’s correlation coefficient between mother’s pBMI and child’s birth weight was 0.21 and denoted insignificant, Figure 1a. After conducting the analysis for gender separately, the trend for male children was significant (Figure 1b) and for female insignificant (Figure 1c).

An overall model, including gender and smoking, indicated a significant impact of pBMI category on child’s birth mass (*p* = 0.01, Effect size = 0.19), Table 2.

The same tendency was noticed for weight at the first month of life. The correlation for the whole group was denoted significant (Figure 2a), however, upon a separate analysis, the trend was significant for male population (Figure 2b), while it was insignificant for female newborns (Figure 2c). The overall model, accounting for gender, smoking, breastfeeding and the type of delivery adjustment, indicated a significant impact of mothers’ pBMI category on child’s mass at the first month of life (*p* = 0.01, Effect size = 0.2), Table 2.

The analysis of child’s weight gain over time among the maternal BMI group (Figure 3) indicated no correlation between time and group after adjusting for gender. The average weight gain per unit time was 1.9185 for underweight, for normal weight was greater by 0.1055, for overweight was lower by −0.0196, for obese was greater by 0.2108 (the coefficients were insignificant).

Having analyzed the prevalence of underweight and overweight in children at 1 and 2 years of age, based on the WHO Z-scores, no children were found underweight, and only one 2-year-old child was found to be obese (a child of an obese mother, who was obese before the pregnancy). Eight children were overweight by the age of one and seven by the age of two. There was no significant correlation between the occurrence of abnormal body weight in children and pBMI of the mother (Table 3).

Based on the centile scales, no weight deficiency was found in any of the assessed children, while excess body weight was measured at both 12 and 24 months of age in five and three children, respectively. Using this scale, there was no relationship between the occurrence of abnormal body weight in children and the pBMI of the mother (Table 3).

Yet, we have obtained significantly different results using the BMI-for-age and sex grids. Six 1-year-old children and four 2-year-olds were found to be underweight, while, 9 and 11 children, respectively, were found to be obese at the same time frames. A significantly higher incidence of obesity was found among 2-year-old children of obese mothers (Table 3).

The general characteristics of mothers and their children according to the maternal GWG category were presented in Table 4. Having qualified in accordance with GWG, 16 participants had adequate, 48 excessive and 18 inadequate GWG. The mean value of pBMI was found to be the highest in the inadequate GWG group, while the lowest in the adequate GWG group.

Having analyzed the impact of maternal GWG category on child’s body mass, a significant trend was found for birth weight (*p* = 0.018, Effect size = 0.41).

No differences in proportion have been found between the occurrence of underweight or obesity in the children of mothers who gained an incorrect amount of weight during the pregnancy (Table 5).

## 4. Discussion

Our research is the first prospective study evaluating the relationship between the body weight of healthy children from birth to 2 years of age, pBMI and maternal gestational weight gain in the Polish population. This early period of life, consisting of the first 24 months of life, is very interesting from the preventive medicine point of view, as even little changes, such the modification of diet and lifestyle make it possible to prevent the negative consequences of obesity and metabolic disorders. In our study, we have found a relationship between pBMI and GWG and the body weight of children at birth and between pBMI and the body weight of children in first month of life. Moreover, we observed a correlation between pBMI and boys’ body weight at birth and at 1 month of age. J. Hu et al., have shown in their study, there may be sex-specific association with gestational weight gain [32]. They observed that greater total GWG was positively associated with male infant’s BMI Z-score at 3 and 6 months. This may have been due to the differences in leptin and testosterone levels (adiposity related hormones) among infants of different sex. In infancy, in accordance to Bouyer et al., greater leptin exposure is likely to suppress appetite and may influence hypothalamic development, limiting the subsequent excess weight gain [33]. However, multiple research has noted a positive correlation between GWG and birth weight regardless the sex of the infant [1,3,34]. These observations indicate in utero overnutrition in children of mothers with pBMI and GWG. The clinical and epidemiological significance of the observed relationships is unclear. A systematic review and a meta-analysis involving 643,902 persons and published in 2012 found that high birth weight may result in becoming overweight in later life [35]. In contrast, Belbasis et al., in an umbrella review of systematic reviews and meta-analyses found no strong evidence for an association between birth weight and disease later in life [36]. In addition, the mean birth weight values in our study were <4000 g, the limit above which there is an increased risk of overweight or obesity in adult life [35].

In the presented study, the association between pBMI and the obesity in children aged 2 (45% of obese children of obese mothers vs. 13% of obese children of mothers with normal body weight) was only significant when BMI-for-age and sex charts were used to diagnose obesity in children. However, using the weight-for age and sex Z-score charts which are recommended by the WHO, the relationships were found to be insignificant. Moreover, we did not find any associations between excessive GWG and childhood obesity. The differences in the prevalence of underweight and obesity in children depending on the meshes used are very interesting. The most similar, but not identical, results were obtained using weight-for age and sex Z-score and centile charts, however, in accordance with the WHO, centile charts should only be used to distinguish the group of children with abnormal body weight who require further diagnostics and supervision. Using BMI for age and sex, underweight and obesity were diagnosed much more frequently. It is true that the WHO does not recommend the use of BMI charts until the age of 2, but they are still used in scientific publications regarding obesity in young children [17,32,37]. This is why the research results presented in numerous publications should be treated with great caution, paying particular attention to the applied criteria for the diagnosis of abnormal growth of young children.

The influence of pBMI on the childrens’ body weight during the first two years of life has been scarcely studied; yet there is some research concerning the relationship between the pBMI, GWG and the body weight of children later in life. Diaz-Rodrigez et al., have evaluated in their prospective study, that included 109 mother-child pairs, the association between pBMI ≥ 25.0 and the percentage fat mass [17]. Moreover, a meta-analysis by Voerman et al., including 162,129 pairs of mothers and children, found an association between pBMI ≥ 30.0 and the incidence of overweight in children from 2 to 18 years of age. The study by Voerman et al., showed that the effect was increasing with age of the offspring [6]. In a retrospective study conducted on a large group of mothers and children in China by Li et al., it was found that the obesity in children at 1 year of age is more common in the offspring of obese mothers than in those with normal body weight (21.4% vs. 17.0%) [38]. A recent American study by Olson et al., reported 36% of children (3.5 to 4.5 years) of obese mothers to be obese, while only 7.8% of children of mothers with normal weight were found to be obese [39]. Studies evaluating the impact of GWG on later life have also revealed that offspring born to mothers with excessive GWG or higher pBMI have a greater risk of overweight and obesity from childhood to adulthood [31,40,41]. A cohort study of 10,226 participants who have participated in the Collaborative Perinatal Project (1959–1972) has reported an increased risk of 48% (95% CI: 1.06, 2.06) of overweight children at 7 years of age of mothers who gained more on weight than recommended in the weight gain recommendations, when compared with the children of mothers who met the weight gain criteria [40,41]. Contrary to the results observed by Li et al., our study did not demonstrate that pre-pregnancy obesity influences obesity in children at 12 months of age [38]. However, it should be noted that in the study conducted by Li et al., the risk of obesity was only increased in children aged 12 months of mothers who had both pre-pregnancy overweight/obesity and excessive GWG. Due to the small study sample size, this type of analysis could not be performed in our research. Moreover, in the Chinese study, maternal obesity was diagnosed when pBMI ≥ 28, while in our study, only when pBMI ≥ 30. As no influence of pBMI on obesity was observed in the first year of life, it can be concluded that most probably the environmental conditions including lifestyle and diet, and not genetic conditions or those related to pregnancy and childbirth exert the greatest impact on the prevalence of obesity among two-year-old children. It can be also assumed, in accordance with Drake and Reynolds, that, according to the “developmental overnutrition hypothesis”, maternal factors (i.e., high glucose, free fatty acids and amino acid concentrations) can cause permanent changes in appetite control, neuroendocrine function and/or energy metabolism in the developing fetus, leading to an increased risk of obesity that manifests only after the first year of life [42].

In this study, we did not observe any influence of pBMI and GWG on breastfeeding duration. Additionally, the percentage of 6-month breastfeeding women was similar among underweight, overweight and obese mothers before pregnancy. Contrary to our results, Chen et al., have shown maternal underweight and obesity to be associated with earlier breastfeeding cessation in Taiwan [43]. Moreover, other authors have demonstrated maternal obesity to have an inverse association with breastfeeding duration [44,45]. Breastfeeding-related behaviors are complex. Maternal obesity may influence multiple perinatal factors including the occurrence of preterm labor and Cesarean section, which are often associated with failure to breastfeed [46,47]. Obesity is often related with socioeconomic status (SES) as in high-income countries, people with higher SES are less likely to be obese. In accordance with Kitsantas et al., as low SES and poorer education status adversely affect breastfeeding, obese women may find it more difficult to continue to breastfeed [48]. The relationship between underweight, which is quite rare in the Western countries, and breastfeeding has not been yet deeply studied in accordance to Huang et al., meaning that further research including different countries and populations is needed [49,50].

The strengths of our study include its prospective nature, a selected group of healthy children and mothers, and their ethnic homogeneity. The results may be different for different populations as weight gain seems to be culturally and ethnically dependent. As demonstrated by other researchers, Black, Hispanic and Asian women tend to gain weight during pregnancy below the current recommendations, while White women are likely to exceed them. Increasing migration and demographic change increase ethnic discrepancies. Due to the variation in gestational weight gain between populations, according to Abarca-Gómez et al., the differences should be taken into consideration while creating international guidelines and the population discrepancies should be further evaluated [51].

The greatest limitation of this study was the small number of patients studied. Despite the fact that it did not reduce the confidence interval (moderate effect size) in which we find significance (type I errors), it may toughen the interpretation of negative results, which can be seen when compared with large cohort studies e.g., those conducted in China [5,32]. As a part of our research, we only used measurements of mothers’ weight at conception, at labor and total gestational weight gain without distinction of GWG in trimesters, only as an exposure factor. Gaining excessive weight at a different time may alter the final outcome as early, mid, and late pregnancy GWG have different associations with fetal growth and later life adiposity. As other researchers have reported, weight gains during the first and second trimester of gestation are most strongly associated with cord blood levels of hormones involved in glycemic control and somatic growth and may mostly correlate with obesity and body adiposity distribution in childhood [32,37]. For organizational reasons, we did not have data on the mother’s nutrition and exercise. Data on alcohol drinking are also unreliable, as women denied drinking alcohol during pregnancy because it is socially unacceptable. We did not also collect data concerning children’s food intake, which could affect the results of weight gain. All babies from 6 months onwards have had other foods introduced in addition to breastfeeding. According to the current nutrition guidelines, these are vegetable soups followed by those with meat, fruit purees and juices, porridges based on breast milk or cow’s milk (depending on whether the woman still has her milk or the baby is given a milk mixture). The differences in the children’s diet only appear in the second year of life. Decreasing prevalence of overweight and obese women of reproductive age especially in perigestational period could reduce the prevalence of obesity in early childhood. In order to increase the percentage of women of a healthy weight, outreach is needed to promote a healthy lifestyle among adolescent and young women. Physical activity and healthy, balanced nutrition should be encouraged. Preconception counseling should be made by both obstetricians and primary care doctors to promote and implement the following BMI guidelines [17,52]. Preventative measures and better education and management of pregnant women should be introduced to diminish potentially significant healthcare related costs of increased odds of adverse fetal and maternal pregnancy outcomes, increased hospital stay and increased inpatient costs [10,53]. As obesity becomes an important issue worldwide, further research on a greater population should be continued to examine the extent of mothers’ body mass and gestational weight gain on infants’ body weight as it is a unique period of time when both mother and her child could potentially benefit at the same time from dietary and lifestyle modifications.

## 5. Conclusions

Poland is one of the countries where the proportion of obese and overweight children is growing very rapidly. Since, according to programming theory, the health status of children and adults is influenced by various events during the fetal and perinatal period, this paper presents the influence of selected factors such as maternal nutritional status, smoking, mode of delivery, antibiotic therapy during pregnancy, feeding children on their body weight from birth to two years of age. Due to the lack of a centralized information system, the results of this study help to obtain statistical data, which allow to draw conclusions on the necessity and direction of early interventions in pregnant women and young children in Poland. It was found that pBMI and GWG are associated with changes of birth weight and weight in neonatal period but do not influence breastfeeding duration, however the impact of these factors on children’s weight and health consequences later in life are unclear. The biggest limitation of this study was the small sample size and the failure to account for weight gain per trimester of pregnancy. Preconception obesity prevention and life style modification of children of obese women is required (modification of diet and physical activity, together with behavioral and social support strategies). The methods used for weight assessment among children aged up to two years of age require standardization. Further research on a larger population should be continued to investigate the extent to which maternal weight and gestational weight gain affect infant weight.

## Figures and Tables

**Figure 1 children-09-01431-f001:**
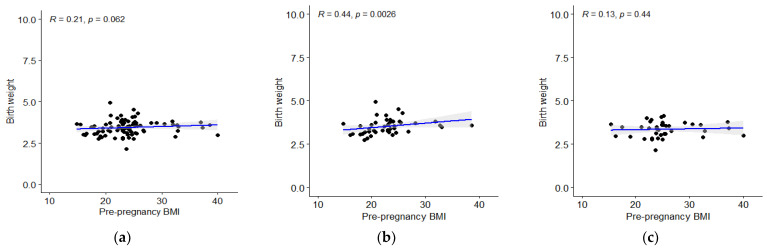
(**a**) The correlation between mother’s pBMI and child’s birth weight; (**b**) correlation between mother’s pBMI and birth weight of male population; (**c**) correlation between mother’s pBMI and weight of female population.

**Figure 2 children-09-01431-f002:**
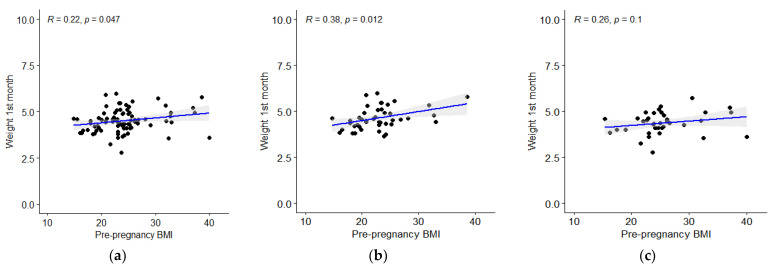
(**a**) Correlation between mother’s pBMI and child’s weight at first month of life; (**b**) correlation between mother’s pBMI and weight of males at first month of life; (**c**) correlation between mother’s pBMI and weight of females at first month of life.

**Figure 3 children-09-01431-f003:**
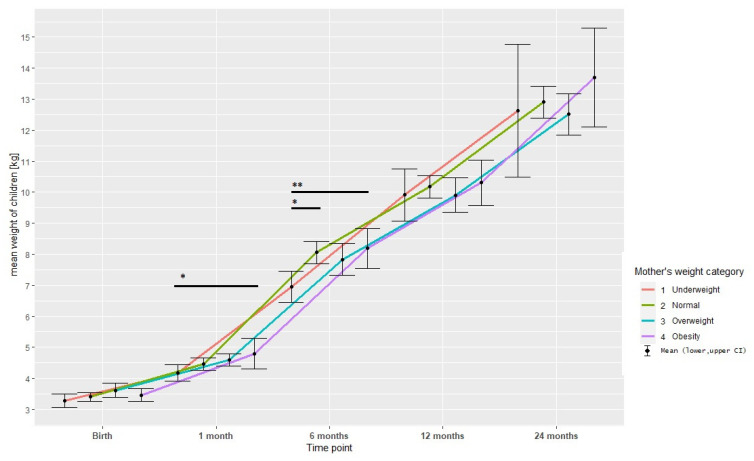
Child’s weight gain over time among the maternal BMI group. Observed differences: 1 month between groups underweight and obesity, 6 months between groups underweight and obesity and underweight and normal. *p* values obtained in post hoc analysis (Dunn’s test): * *p* < 0.05, ** *p* < 0.001.

**Table 1 children-09-01431-t001:** The prevalence of underweight, overweight and obesity in the study group according to the different classifications.

Child’s Weight Status	Centile Charts	12 Month		24 Month	
		Male	Female	Male	Female
Underweight	Z-score (kg): <−2	<7.7	<7.0	<9.7	<9.0
	Centile charts (kg): <2.3 centile (abnormal growth)	<7.8	<7.1	<9.8	<9.2
	BMI (kg/m^2^): <3 percentile	<14.5	<13.9	<13.7	<13.2
Overweight	Z-score (kg): 2–3	12.0–13.3	11.5–13.1	15.3–17.2	14.8–17.0
Obesity	Z-score (kg): >3	>13.3	>13.1	>17.2	>17.0
	Centile charts (kg): >97.7 centile (abnormal growth+)	>11.8	>11.3	>15.1	>14.6
	BMI (kg/m^2^): >97 percentile	>19.6	>19.4	>18.3	>18.2

**Table 2 children-09-01431-t002:** The characteristics of study participants among 83 mother-infant pairs according to pBMI weight categories in Szczecin, Poland.

	pBMI (kg/m^2^)				*p* Adjusted
	<18.5	18.5–24.9	25–29.9	≥30	
Number of subjects; n (%)	9 (11)	46 (55)	17 (20)	11 (13)	
Maternal characteristics					
pBMI(kg/m^2^)	16.73 (1.23)	22.34 (1.81)	25.94 (1.17)	34.38 (3.21)	NA
Mass gain(kg)	16.13 (4.94)	17.13 (6.04)	16.3 (5.51)	7.12 (7.7)	0.0004
Breastfeeding (weeks)	17.75 (17.35)	26.05 (26.01)	28.03 (22.75)	24.91 (24.55)	0.799
Breastfeeding at 6th month; n (%)	2 (25)	19 (41.3)	9 (52.9)	5 (45.4)	0.875
Smoking+; n (%)	2 (22)	13 (28.2)	1 (6)	2 (18)	0.3973
ATB+; n (%)	2 (22)	12 (26)	7 (41)	5 (45)	0.441
GBs+; n (%)	1 (12.5)	10 (29.4)	5 (38.5)	4 (40)	0.614
Child characteristics					
Delivery C, n (%)	5 (55.6)	30 (65.2)	11 (64.7)	6 (54.5)	0.875
Gender male; n (%)	6 (66.7)	28 (60.8)	6 (35.3)	4 (36.4)	0.315
Weight [kg]					
Birth	3.29 (0.29)	3.41 (0.49)	3.61 (0.45)	3.47 (0.3)	0.010
1 month	4.17 (0.34)	4.47 (0.65)	4.59 (0.39)	4.81 (0.71)	0.010
6 months	6.96 (0.54)	8.06 (1.18)	7.83 (0.96)	8.19 (0.9)	0.315
12 months	9.91 (0.91)	10.17 (1.18)	9.91 (1.04)	10.31 (1.02)	0.491
24 months	12.63 (2.05)	12.91 (1.59)	12.51 (1.21)	13.7 (2.23)	0.251

BMI—body mass index, n—size of the group, C—cesarean section, +—presence of the factor, *p*—*p* value, NA—not applicable; ATB—antibiotic use during pregnancy and childbirth, GBs—vaginal colonization with *Streptococcus agalactiae.* For quantitative variables group’s mean and (SD) is presented. *p* values were obtained by multiple linear regression, after FDR correction.

**Table 3 children-09-01431-t003:** Occurrence of abnormal body weight in children according to the pBMI category.

**Classification by Z-Score, 12 Months**					
**Child’s Weight Category (n)**	**pBMI Category**				*p* Value
	**Underweight**	**Normal**	**Overweight**	**Obesity**	
Underweight	0	0	0	0	0.63
Normal	7	16	15	9	
Overweight	0	6	1	1	
Obesity	0	0	0	0	
**Classification by Z-score, 24 months**					
**Child’s Weight Category (n)**	**pBMI category**				*p* value
	**Underweight**	**Normal**	**Overweight**	**Obesity**	
Underweight	0	0	0	0	0.15
Normal	5	35	15	7	
Overweight	1	4	0	2	
Obesity	0	0	0	1	
**Classification by centile charts: weight-for-age, 12 months**					
**Child’s weight status (n)**	**pBMI category**				*p* value
	**Underweight**	**Normal**	**Overweight**	**Obesity**	
abnormal growth −	0	0	0	0	0.527
Normal	5	21	5	2	
abnormal growth +	0	4	0	1	
**Classification by centile charts: weight-for-age, 24 months**					
**Child’s weight status (n)**	**pBMI category**				*p* value
	**Underweight**	**Normal**	**Overweight**	**Obesity**	
abnormal growth −	0	0	0	0	0.252
Normal	5	21	5	2	
abnormal growth +	1	1	0	1	
**Classification by centile charts: BMI-for-age, 12 months**					
**Child’s BMI category (n)**	**pBMI category**				*p* value
	**Underweight**	**Normal**	**Overweight**	**Obesity**	
Underweight	1	3	0	2	0.582
Other groups †	7	41	10	6	
Other groups *	8	36	10	7	0.241
Obesity	0	8	0	1	
**Classification by centile charts: BMI-for-age, 24 months**					
**Child’s BMI category (n)**	**pBMI category**				*p* value
	**Underweight**	**Normal**	**Overweight**	**Obesity**	
Underweight	1	2	1	0	0.745
Other groups †	3	28	12	9	
Other groups *	4	24	13	4	0.04
Obesity	0	6	0	5	

* Other groups: normal weight, overweight, obesity. − reduction of weight. + addition of weight. † Other groups: underweight, normal weight, overweight.

**Table 4 children-09-01431-t004:** The general characteristics of mothers and their children according to the maternal GWG category.

	GWG Category			*p* Adjusted
	Inadequate	Adequate	Excessive	
Number of subjects; n (%)	18(22)	16(19.5)	48(58.5)	
**Maternal characteristics**				
BMI gain (kg/m^2^)	2.28(1.68)	4.83(0.93)	7.14(1.64)	NA
pBMI (kg/m^2^)	25.17(7.74)	22.58(3.72)	24.20(4.24)	1
Natural breastfeeding; n (%)	11(61.11)	11(68.75)	34(70.83)	1
Breastfeeding (weeks)	30.00(30.76)	20.09(24.62)	26.16(21.6)	1
Breastfeeding at 6th month	7(41.2)	5(31.2)	23(47.9)	1
Smoking+; n (%)	4(22.22)	3(18.75)	10(20.83)	1
**Child characteristics**				
Delivery C, n (%)	8(50)	8(44.44)	35(72.91)	0.165
Gender male; n (%)	8(44.44)	11(68.75)	25(52)	0.507
Weight (kg)				
Birth	3.19(0.41)	3.38(0.3)	3.57(0.45)	0.018
1 month	4.28(0.75)	4.57(0.44)	4.59(0.56)	0.081
6 months	7.45(1.21)	8.01(0.62)	8.07(1.14)	0.375
12 months	9.69(1.07)	10.44(1.01)	10.17(1.1)	0.307
24 months	12.20(1.63)	12.54(1.26)	13.27(1.69)	0.288

C—cesarean section, NA—not applicable for quantitative variables group’s mean and (SD) were presented. GWG categories: inadequate (1): <12.5 kg (pre-pregnancy BMI < 18.5 kg/m^2^), <11.5 kg (BMI 18.5–23.9 kg/m^2^), <7 kg (BMI 24.0–27.9 kg/m^2^), and <5 kg (BMI > 28 kg/m^2^); adequate (1): 12.5–18 kg (BMI < 18.5 kg/m^2^), 11.5–16 kg (BMI 18.5–23.9 kg/m^2^), 7–11.5 kg (BMI 24.0–27.9 kg/m^2^), and 5–9 kg (BMI > 28 kg/m^2^); excessive (1): >18 kg (BMI < 18.5 kg/m^2^), >16 kg (BMI 18.5–23.9 kg/m^2^), >11.5 kg (BMI 24.0–27.9 kg/m^2^), and >9 kg (BMI > 28 kg/m^2^), according to the 2009 Institute of Medicine/National Research Council GWG recommendations [31]. Smoking+ definition: a woman was considered to have smoked cigarettes if she did so daily throughout her pregnancy. Women who smoked during pregnancy also smoked before pregnancy. Women who had given up smoking by 12 Hbd were not classified as smokers. In addition, information on the number of cigarettes smoked per day was collected, but additional subgroups were dropped due to the small size of the study group.

**Table 5 children-09-01431-t005:** Occurrence of abnormal body weight in children according to the maternal GWG category.

**Classification by Z-Score, 12 Months**				
**Child’s Weight Category (n)**	**Maternal GWG Category**			*p* Value
	**Inadequate**	**Adequate**	**Excessive**	
Underweight	0	0	0	0.87
Normal	16	12	39	
Overweight	1	2	5	
Obesity	0	0	0	
**Classification by Z-score, 24 months**				
**Child’s weight category (n)**	**Maternal GWG category**			*p* value
	**Inadequate**	**Adequate**	**Excessive**	
Underweight	0	0	0	0.08
Normal	15	11	36	
Overweight	0	0	7	
Obesity	1	0	0	
**Classification by centile charts: weight-for-age, 12 months**				
**Child’s weight status (n)**	**Maternal GWG category**			*p* value
	**Inadequate**	**Adequate**	**Excessive**	
abnormal growth −	0	0	0	1
Normal	6	8	19	
abnormal growth+	1	1	3	
**Classification by centile charts: weight-for-age, 24 months**				
**Child’s weight status (n)**	**Maternal GWG category**			*p* value
	**Inadequate**	**Adequate**	**Excessive**	
abnormal growth −	0	0	0	0.756
Normal	6	7	17	
abnormal growth +	0	0	3	
**Classification by centile charts: BMI-for-age, 12 months**				
**Child’s BMI category (n)**	**Maternal GWG category**			*p* value
	**Inadequate**	**Adequate**	**Excessive**	
Underweight	4	0	4	0.76
Other groups *	10	12	40	
Other groups †	13	11	38	0.159
Obesity	1	1	6	
**Classification by centile charts: BMI-for-age, 24 months**				
**Child’s weight status (n)**	**Maternal GWG category**			*p* value
	**Inadequate**	**Adequate**	**Excessive**	
Underweight	1	1	2	0.76
Other groups *	11	8	33	
Other groups †	9	8	28	1
Obesity	3	1	7	

* Other groups: normal weight, overweight, obesity. † Other groups: underweight, normal weight, overweight. − small weigh for age increase, + too big weight for age increase.

## Data Availability

The data presented in this study are available on request from the corresponding author. The data are not publicly available due to ethical restrictions.

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
