# Peer review of "Analysis of the Influence of Pre-Pregnancy BMI and Weight Gain during Pregnancy on the Weight of Healthy Children during the First 2 Years of Life: A Prospective Study"

_children, 2022, doi:10.3390/children9101431_

Round 1
Reviewer 1 Report
Dear authors,
The article presents an interesting topic, however I leave some suggestions for improvement:
- The abstract should mention the type of article, the method of data collection, as well as the type of analysis applied;
- I also suggest that there should be a short theoretical framework on the subject (literature review) after the introduction. Besides enriching the article more, it would increase the interest of those who read it;
- It presents very good results, however the conclusion is very poor, which should be further developed: present proposals for future studies, limitations and the contributions of the article.
- I also leave a note for the number of authors...
Author Response
Rev #1
The abstract should mention the type of article, the method of data collection, as well as the type of analysis applied;
Response
Thank you for this comment. The abstract was revised accordingly.
I also suggest that there should be a short theoretical framework on the subject (literature review) after the introduction. Besides enriching the article more, it would increase the interest of those who read it;
Response
Thank you for this suggestion. We have added the sentence "In addition, there are several observations (including a meta-analysis) involving the effect of pBMI on children's body weight later in life" to the Introduction section. In addition, the discussion describes in great detail the observations available in the literature on the relationship between maternal and child body weight along with the proposed pathomechanisms.
It presents very good results, however the conclusion is very poor, which should be further developed: present proposals for future studies, limitations and the contributions of the article.
Response
Thank you for these comments, the manuscript was amended accordingly.
I also leave a note for the number of authors...
Response
It should be taken into account that there is no centralised medical information system in Poland, so all data must be collected 'manually' - by contacting the patient. Considering the number of subjects, frequency of contacts and duration of the study, the number of authors of this paper does not seem excessive.
Reviewer 2 Report
The authors have aimed to assess the impact of underweight, overweight, and obesity of women before pregnancy and gestational weight gain on the weight of children and breastfeeding duration from birth to 2 years of age. Overall, the authors have examined a very interesting research question and very timely with the changing environments. It is very nice to see data on changes in child weight status across the first 2 years of life which is considered to play a critical role in growth and development. Improvement needs to be made throughout the paper to address wording and definition issues. These changes will improve readability of the manuscript and help with interpretation of findings.
Here are detailed comments:
Introduction
1. Line 49: Are the authors referring to “Fetal Programming”?
2. Line 57: Please provide some examples of “short term lifestyle modifications”.
3. The authors should consider including a stronger reasoning for performing a study using a Polish population. What was the motivation for looking into a Polish population? Is it only because there is a knowledge gap in this area among Polish population or are Polish women at higher risk for other factors?
4. Line 71: Change “before pregnancy” to “pre-pregnancy”.
5. Consider rewording the objectives, it is hard to read the way it is written.
Methods
1. How was sample size determined?
2. Please specify the inclusion criteria more clearly.
3. Line 96: how was antibiotic treatment determined?
4. Lines 99-100: Please include a reference. Was WHO definitions used to classify BMI?
5. Line 102: the word “excessive” comes twice. Should it be “inadequate, normal, and excessive? Please provide a definition for these categories.
6. Lines 107-108: Include a reference.
7. How was “child mass” defined?
8. Lines 121-122: how were the covariates selected for adjustment?
9. Were models adjusted for maternal age?
1. Is there any data available on maternal diet or other lifestyle factors such as physical activity or alcohol intake? Considering some of these factors in the models will help strengthen the findings.
Results
1. Line 136: how was cigarette smoking defined? Is there information available on the number of cigarettes smoked or frequency or duration of smoking? Previous literature has suggested associations between smoking and low birth weight. It would be helpful to know how smoking was entered in the model.
2. Line 136: Please be consistent. Either use “pre-pregnancy BMI” or “BMI before pregnancy”
3. Line 137: Were women weighed before labor to determine BMI before labor? If so, then how was weight during labor measured? Was it at the start of labor or anytime after onset of labor?
4. Line 140: Instead of normal weight, maybe can explain using normal BMI otherwise please define the normal weight range.
5. Line 143: Please provide a definition for determining breastfeeding in infants in the methods. Table 2 indicated “weeks”. Since breastfeeding is part of the objectives examined, it will help to have a clearer definition in the methods.
6. Which statistical test was used to obtain the p-value given in Table 2? Please add a footnote with these details.
7. Table 2 lists “Delivery C”, please add a definition in the footnote.
8. In Figure 1b and subsequent figures were gender differences have been examined, please use “male” or “female” instead of “son” or “daughter” to maintain consistency throughout the paper.
9. Figure 3 looks really nice. It would be better to rename the y-axis to read “mean weight of children”
1. Table 3: it was very hard to understand that these numbers meant the “number of children” in each category. Please add “n” somewhere so it is easier to understand.
Discussion
1. No comments.
Author Response
Introduction
- Line 49: Are the authors referring to “Fetal Programming”?
Response
Thank you for this comment. Information on fetal programming has been included in the manuscript.
- Line 57: Please provide some examples of “short term lifestyle modifications”.
Response
Thank you for this comment. We have changed the term 'short term lifestyle modifications' to 'lifestyle modifications', which better reflects the meaning of the sentence. We have also added in brackets the types of the lifestyle modification (modification of diet and physical activity, together with behavioural and social support strategies). One example of such modification is limiting calorie intake at ~20 kcal/kg body weight; increasing physical activity, with a goal of 30 minutes of activity most days of the week; daily recording of food intake, activity and weight; development of control techniques; strategies related to the home environment (e.g. 'cleaning' cupboards, posting visual cues). (https://lifemoms.bsc.gwu.edu/documents/8058629/10071547/LIFE-Moms+Synopses.pdf/de5a0dca-3da5-4a8f-af7d-31f2eee8c77a accessed 06-09-2022)
- The authors should consider including a stronger reasoning for performing a study using a Polish population. What was the motivation for looking into a Polish population? Is it only because there is a knowledge gap in this area among Polish population or are Polish women at higher risk for other factors?
Response
Thank you for this comment. We added the following paragraph in Introduction: According to official data (https://stat.gov.pl/obszary-tematyczne/zdrowie/zdrowie/odsetek-osob-w-wieku-powyzej-15-lat-wedlug-indeksu-masy-ciala-bmi,23,1.html dotsęp 05-09-2022), between 2009 and 2019, the percentage of overweight (BMI 25 - 29.9) among women aged 20 - 39 increased from 25.8 to 31.3. This means that more than 30% of women of childbearing age are overweight. This is therefore a significant social and health problem. In Poland, there is no central medical data collection system, so epidemiological studies are based on data obtained from individual cohorts.
- Line 71: Change “before pregnancy” to “pre-pregnancy”.
Response
Thank you, we have unified the terminology throughout the manuscript.
- Consider rewording the objectives, it is hard to read the way it is written.
Response
We agreed with this comment and corrected the aims accordingly.
Methods
- How was sample size determined?
Response
We’ve calculated minimum sample size (P = 0.8, α = 0.05) for Cohen’s f^2 small, medium and large effect size. Calculated minimum sample size to detect small effects was 61 subjects, our study included 83 subjects. We included this information in the manuscript.
- Please specify the inclusion criteria more clearly.
Response
We amended text as follows: “The inclusion criteria: healthy, term newborns born to healthy mothers in good condition; all qualified newborns after birth were rated above 7 points on the Apgar scale after 3 min of life, and the pH of the umbilical cord blood was > 7.2. Mothers who had a history infection during pregnancy were not excluded from the study”.
- Line 96: how was antibiotic treatment determined?
Response
Antibiotic therapy was recorded by using the Pregnancy Chart and the history collected from the mothers. Apart from the name of the antibiotic, we do not have detailed information on the doses used and the length of therapy. We have included this information in the manuscript.
- Lines 99-100: Please include a reference. Was WHO definitions used to classify BMI?
Response:
The classification for children, followed the WHO guidelines, for mothers followed the CDC guidelines. The WHO Child Growth Standards [WWW Document], n.d. URL https://www.who.int/tools/child-growth-standards/standards (accessed 2.23.22). We have included this information in the manuscript.
- Line 102: the word “excessive” comes twice. Should it be “inadequate, normal, and excessive? Please provide a definition for these categories.
Response
Thank you for this comment. The mistake was corrected.
- Lines 107-108: Include a reference.
Response
Reference was included.
- How was “child mass” defined?
Response
Child's weight status was determined by the criteria shown in Table 1. Analyses of occurrence of abnormal body weight in children according to the maternal pBMI category were based on all the classifications. We found that the association between pBMI and the obesity in children aged 2 (45% of obese children of obese mothers vs 13% of obese children of mothers with normal body weight) was only significant when BMI-for-age and sex charts were used to diagnose obesity in children. However, using the weight-for age and sex Z-score charts which are recommended by the WHO, the relationships were found to be insignificant. Therefore, we formulated the conclusion that: “The methods used for weight assessment among children aged up to two years of age require standardization”.
- Lines 121-122: how were the covariates selected for adjustment?
Response
Smoking, gender, breastfeeding and delivery type are known to affect children’s body weight. Therefore we selected these covariates for adjustment. The following references were included in the manuscript:
Schnurr, T.M., Ängquist, L., Nøhr, E.A. et al. Smoking during pregnancy is associated with child overweight independent of maternal pre-pregnancy BMI and genetic predisposition to adiposity. Sci Rep 12, 3135 (2022). https://doi.org/10.1038/s41598-022-07122-6
Ralphs, E., Pembrey, L., West, J. et al. Association between mode of delivery and body mass index at 4-5 years in White British and Pakistani children: the Born in Bradford birth cohort. BMC Public Health 21, 987 (2021). https://doi.org/10.1186/s12889-021-11009-y
Scott, J.A., Ng, S.Y. & Cobiac, L. The relationship between breastfeeding and weight status in a national sample of Australian children and adolescents. BMC Public Health 12, 107 (2012). https://doi.org/10.1186/1471-2458-12-107
Shah B, Tombeau Cost K, Fuller A, et alSex and gender differences in childhood obesity: contributing to the research agendaBMJ Nutrition, Prevention & Health 2020;bmjnph-2020-000074. doi: 10.1136/bmjnph-2020-000074
- Were models adjusted for maternal age?
Response
The mother's age was not included in the analysis, because this factor is not widely recognised as having a significant impact on children's body weight.
- 10. Is there any data available on maternal diet or other lifestyle factors such as physical activity or alcohol intake? Considering some of these factors in the models will help strengthen the findings.
Response
Thank you for this important comment. Gathering reliable information on diet requires the respondent to keep a food diary and complete a multi-page questionnaire. As the pre-qualification of mothers for the study took place at the time the woman reported for labour and the final qualification after the birth of a healthy newborn, it was not possible to obtain reliable information on diet. All women denied drinking alcohol, but given the shame of Polish pregnant women before admitting this fact, these data can certainly not be regarded as fully reliable. As there are no validated questionnaires for pregnant women in Poland regarding their physical exercise, no information was collected on this topic. We have included this information in the study limitations.
Results
- Line 136: how was cigarette smoking defined? Is there information available on the number of cigarettes smoked or frequency or duration of smoking? Previous literature has suggested associations between smoking and low birth weight. It would be helpful to know how smoking was entered in the model.
Response
A woman was considered to have smoked cigarettes if she did so daily throughout her pregnancy. Women who smoked during pregnancy also smoked before pregnancy. Women who had given up smoking by 12 Hbd were not classified as smokers. In addition, information on the number of cigarettes smoked per day was collected, but additional subgroups were dropped due to the small size of the study group. This information was added in the description of table 4.
- Line 136: Please be consistent. Either use “pre-pregnancy BMI” or “BMI before pregnancy”
Response
Thank you for your attention and we apologize for this shortcoming, which we have tried to correct throughout the manuscript.
- Line 137: Were women weighed before labor to determine BMI before labor? If so, then how was weight during labor measured? Was it at the start of labor or anytime after onset of labor?
Response
Women were weighed prior to delivery when they reported to the hospital.
- Line 140: Instead of normal weight, maybe can explain using normal BMI otherwise please define the normal weight range.
Response
Thank you. We did corrections.
- Line 143: Please provide a definition for determining breastfeeding in infants in the methods. Table 2 indicated “weeks”. Since breastfeeding is part of the objectives examined, it will help to have a clearer definition in the methods.
Response
We have amended the methodology by stating: “The breastfeeding period took into account exclusive and partial breastfeeding times”.
- Which statistical test was used to obtain the p-value given in Table 2? Please add a footnote with these details.
Response
We used multiple linear regression, adjusted by smoking, gender, breastfeeding and delivery type.
- Table 2 lists “Delivery C”, please add a definition in the footnote.
Response
Thank you, the information was added.
- In Figure 1b and subsequent figures were gender differences have been examined, please use “male” or “female” instead of “son” or “daughter” to maintain consistency throughout the paper.
Response
Thank you, we made this correction.
- Figure 3 looks really nice. It would be better to rename the y-axis to read “mean weight of children”
Response
We renamed the y-axis.
- Table 3: it was very hard to understand that these numbers meant the “number of children” in each category. Please add “n” somewhere so it is easier to understand.
Response
Thank you for this valuable comment. The Table 3 was prepared de novo.
Reviewer 3 Report
Obesity is a disease with a very high prevalence in the world, affecting all ages and latitudes. Pregestational overweight and obesity affect the nutritional status and the development of other complications in newborns in the short and long term.
The following comments are sent to the authors for their consideration:
1. What is new in this study vs what is reported in the literature?
2. Was the difference in length evaluated? It has been seen that height is also modified according to the nutritional status of the mother.
3. What proposals could be made to improve the nutritional status of pre-pregnancy mothers or weight gain during pregnancy?
4. What do you mean by naturally vs. Artificially?
5. Who collected the information? Who weighed and measured the children? Was it a standardized professional?
6. Was it asked about exclusive breastfeeding?
7. In limitations, is it important to describe whether the children's food intake could affect the results of weight gain? And not just the breastfeeding factor.
8. In the methods section, it is indicated that the Z-scores were obtained according to the WHO references, however, in Table 1, in addition to the z-score, percentiles are shown, which duplicates the information. It is suggested to leave only the z-scores or to support in methodology why to leave the two classifications.
9. Table 2 shows the pre-pregnancy BMI, however, in the category of overweight women, the mean BMI was 25.94 with a standard deviation of 1.17, if so they are not well classified, so there would be mothers who fall into normal weight.
10. In table 2 it is suggested to place the statistical test used at the bottom of the table and describe it in the statistical analysis section (Anova? or Kruskal-Wallis?)
11. Researchers are suggested to put together the correlation tables in such a way that they do not take up so much space in the manuscript. What is the point of assessing correlations separately between boys and girls?
12. It is suggested that in Figure 3 the confidence intervals be obtained and the groups that are different be identified at the different times. The Y axis is from 2 or 3 kg.
13. Why are the numbers of occurrences shown in tables 3 and 5 according to the nutritional status of the child and the mother? Little is understood these tables. It is suggested to obtain differences between the study groups with the weight variable continuously at different times. The authors are suggested to redo the analyses.
Author Response
- What is new in this study vs what is reported in the literature?
Response
We added in Conclusions the following part: Poland is one of the countries where the proportion of obese and overweight children is growing very rapidly. Since, according to programming theory, the health status of children, adults is influenced by various events during the fetal and perinatal period, this paper presents the influence of selected factors such as maternal nutritional status, smoking, mode of delivery, antibiotic therapy during pregnancy, feeding children on their body weight from birth to two years of age. Due to the lack of a centralized information system, results of this study help to obtain statistical data, which allow to draw conclusions on the necessity and direction of early interventions in pregnant women and young children in Poland.
- Was the difference in length evaluated? It has been seen that height is also modified according to the nutritional status of the mother.
Response
In Poland, the length of newborn babies at birth is measured using a centimetre, taking into account the curvature of the baby's body. This method is subject to error, for example due to the presence of a forehead in some of these babies. This method is also used in GP surgeries for small children until they reach a length of 100 cm. As some surgeries use standardised measures to measure children's length, it is not possible to interpret this parameter reliably in all children together, hence the decision to omit interpreting body length in the study, despite having such data.
- What proposals could be made to improve the nutritional status of pre-pregnancy mothers or weight gain during pregnancy?
Response
In conclusions we propose: modification of diet and physical activity, together with behavioural and social support strategies.
- What do you mean by naturally vs. Artificially?
Response
We added the following information in section Methods: Natural feeding is the feeding of breast milk - either directly from the breast or pumped into a bottle. Artificial feeding involves giving the baby a milk mixture
- Who collected the information? Who weighed and measured the children? Was it a standardized professional?
Response
Information was collected by a neonatologist and an obstetrician. Measurements of newborn babies in the Hospital were carried out by a qualified midwife with a specialisation in neonatology. Measurements of children in the following months of life were taken in the offices of the PCPs by the nurses working there. The results of the measurements were entered in the Child Health Booklet and confirmed with the stamp of the person taking the measurements. Weighing of babies was always done on scales designed for this purpose and certified for reliability. The neonatologist involved in the study wrote them down during a home visit, during which he collected a sample of the child's stool from the parents, which was the subject of another study.
- 6. Was it asked about exclusive breastfeeding?
Response
We don’t have this data.
- In limitations, is it important to describe whether the children's food intake could affect the results of weight gain? And not just the breastfeeding factor.
Response
We added in limitations the following statement: “We did not also collect data concerning children's food intake, which could affect the results of weight gain. All babies from 6 months onwards have had other foods introduced in addition to breastfeeding. According to the current nutrition guidelines, these are vegetable soups, followed by those with meat, fruit purees and juices, porridges based on breast milk or cow's milk (depending on whether the woman still has her milk or the baby is given a milk mixture). The differences in the children's diet only appear in the second year of life”.
- In the methods section, it is indicated that the Z-scores were obtained according to the WHO references, however, in Table 1, in addition to the z-score, percentiles are shown, which duplicates the information. It is suggested to leave only the z-scores or to support in methodology why to leave the two classifications.
Response
The interesting part of the study was to demonstrate the importance of choosing an appropriate method for classifying children's nutritional status. Table 1 shows that there are differences in children's body weight that entitle a child to be considered malnourished, overweight or obese depending on the method of classification chosen. Analyses of occurrence of abnormal body weight in children according to the maternal pBMI category were based on all the classifications. We found that the association between pBMI and the obesity in children aged 2 (45% of obese children of obese mothers vs 13% of obese children of mothers with normal body weight) was only significant when BMI-for-age and sex charts were used to diagnose obesity in children. However, using the weight-for age and sex Z-score charts which are recommended by the WHO, the relationships were found to be insignificant. Therefore, we formulated the conclusion that: “The methods used for weight assessment among children aged up to two years of age require standardization”.
- Table 2 shows the pre-pregnancy BMI, however, in the category of overweight women, the mean BMI was 25.94 with a standard deviation of 1.17, if so they are not well classified, so there would be mothers who fall into normal weight.
Response
Fully uncharacterized data may give this impression, however the following table shows that the qualification of the study participants was carried out correctly.
- In table 2 it is suggested to place the statistical test used at the bottom of the table and describe it in the statistical analysis section (Anova? or Kruskal-Wallis?)
Response
We used; multiple linear regression, adjusted by smoking, gender, breastfeeding and delivery type. The sentence: P. values were obtained by multiple linear regression, after FDR correction was added.
- Researchers are suggested to put together the correlation tables in such a way that they do not take up so much space in the manuscript. What is the point of assessing correlations separately between boys and girls?
Response
We made changes according to Reviewer’s suggestion. There are studies which show this correlation in boys. Please see discussion and ref. 31 and 32.
- It is suggested that in Figure 3 the confidence intervals be obtained and the groups that are different be identified at the different times. The Y axis is from 2 or 3 kg.
Response
Thank you for this comment. Fig. 3 was changed.
- Why are the numbers of occurrences shown in tables 3 and 5 according to the nutritional status of the child and the mother? Little is understood these tables. It is suggested to obtain differences between the study groups with the weight variable continuously at different times. The authors are suggested to redo the analyses.
Response
Thank you for this comment. Readability of table 3 and 5 was very low, therefore we prepared them again.
Round 2
Reviewer 2 Report
The authors have done a nice job of incorporating the comments. A few minor comments:
1. In the beginning of the introduction, please first use pre-pregnancy BMI and then in brackets mention pBMI. Then use "pBMI" throughout t the paper. The abbreviation was not defined in the beginning.
2. Towards the end of the introduction, better to keep "our hypothesis" instead of "research hypothesis". In the hypothesis instead of using greater try a different word such as "higher".
3. Page 7: in the beginning before figure 3, it is fine to only say "underweight, normal weight, overweight, and obese" instead of using category before each of these words.
Author Response
Rev #2
- In the beginning of the introduction, please first use pre-pregnancy BMI and then in brackets mention pBMI. Then use "pBMI" throughout t the paper. The abbreviation was not defined in the beginning.
Thank you for your notice. The correction was made.
- Towards the end of the introduction, better to keep "our hypothesis" instead of "research hypothesis". In the hypothesis instead of using greater try a different word such as "higher".
We agree with this suggestion and followed it in the manuscript
- Page 7: in the beginning before figure 3, it is fine to only say "underweight, normal weight, overweight, and obese" instead of using category before each of these words.
Thank you for this comment. We made corrections, please see the improved version of the manuscript
Reviewer 3 Report
I would like to thank the authors for their efforts to improve the manuscript.
Some rearrangements and clarifications have been made to the text. The paper is now somewhat clearer.
Some minor changes should be done.
In graph 3 it is recommended to remove the boxes and leave only the confidence interval line in order to make it easier for the reader to read.
Author Response
- In graph 3 it is recommended to remove the boxes and leave only the confidence interval line in order to make it easier for the reader to read.
We agree with your suggestion. Please see the improved version of Figure 3.
We have also made some minor improvements in accordance with the other reviewers.